# Ceramic Cracks Segmentation with Deep Learning

Gerivan Santos Junior * , Janderson Ferreira, Cristian Millán-Arias , Ramiro Daniel, Alberto Casado Junior and Bruno J. T. Fernandes

Polytechnic School of Pernambuco, University of Pernambuco, Recife 50720-001, Brazil; jrb@ecomp.poli.br (J.F.); ccma@ecomp.poli.br (C.M.-A.); rdbr@poli.br (R.D.); acasado@poli.br (A.C.J.); bjtf@ecomp.poli.br (B.J.T.F.)
* Correspondence: gcsj@ecomp.poli.br

**Abstract:** Cracks are pathologies whose appearance in ceramic tiles can cause various damages due to the coating system losing water tightness and impermeability functions. Besides, the detachment of a ceramic plate, exposing the building structure, can still reach people who move around the building. Manual inspection is the most common method for addressing this problem. However, it depends on the knowledge and experience of those who perform the analysis and demands a long time and a high cost to map the entire area. This work focuses on automated optical inspection to find faults in ceramic tiles performing the segmentation of cracks in ceramic images using deep learning to segment these defects. We propose an architecture for segmenting cracks in facades with Deep Learning that includes an image pre-processing step. We also propose the Ceramic Crack Database, a set of images to segment defects in ceramic tiles. The proposed model can adequately identify the crack even when it is close to or within the grout.

**Keywords:** deep learning; segmentation; ceramics; cracks; image





## 1. Introduction

In civil construction, buildings must be able to withstand the action of degradation agents for a predetermined or predicted time [1]. The building's facades include the cladding system that serves to protect the building from external degradation agents, in addition to providing functional and aesthetic comfort to its users [2]. Pathological manifestations are common at these points, and they occur more frequently in ceramic materials, which are used on a large scale in buildings. Besides, these manifestations arise in other types of materials, such as mortar and stone. They can be related to several factors such as excessive load, humidity variation, thermal variation, biological agents, material incompatibility, and atmospheric agents [3]. These manifestations compromise the essential function of protection, which aims to protect the coated surfaces against the agents that cause deterioration that can present themselves in different ways. Thus, the consequences can range from aesthetic problems or performance of coating to risks of accidents with people, substantially aggravated by the height of the buildings [4].

The main types of pathological manifestations associated with ceramic facade coverings are cracks, efflorescence, detachment, and those resulting from biological processes. Among these, the fissure is the most found in the literature since it compromises the building safety, puts at risk the people that travel around it, and presents a more critical aesthetic aspect [3,5–7].

A fissure's main characteristic is the rupture appearance on the ceramic plate surface or body, causing the loss of the facade's integrity and uncovers some of its components, the plates, or joints. When the fissure happens, a detachment of the substrate plate is generated [4].

Image processing techniques (IPTs) are currently applied in civil engineering for images collected from inspections. These techniques emerged to detect cracks in the civil infrastructure, partially reducing the work done by human beings, and used several image

processing techniques to extract characteristics of cracks in the surfaces of the images [8]. However, many structure analyses and inspections are carried out manually, and this requires a lot of knowledge, experience, and time from those who will perform this activity, thus making the activity long and time-consuming.

Automatic crack detection is essential in places that are difficult to access due to height or scale, to avoid exposing people to dangerous situations, and to speed up the inspection process [9]. On the other hand, applying procedures takes time due to the complexity of the work performed, including the installation of scaffolding, observation of a large area, and even the use of elevators or Bosun's chair.

Therefore, to create an automatic crack detection solution in ceramics, we focus on an image segmentation methodology. Such methodology includes a pre-processing step because many ceramics have textures in opposition to concrete that has linearity in its texture. The proposed methodology also includes a deep learning model to solve the problem of crack detection in ceramics, which in the future may be coupled to drones to carry out these inspections in a less manual, faster, and less dependent on human action with specific knowledge for the area. This solution also allows the location of the crack to be identified by means of the segmented image, since the crack is segmented, showing its exact location, because in facade inspections, in addition to identifying that there is a crack, which other works do through classification, it is important to know where it is located. We can generate an overlay of the images, highlighting where the crack is for future analysis through segmentation. We created a database to implement the segmentation models that contain images of defective ceramic plates and the basic crack truth in each image.

In summary, this work has as a novelty the detection of cracks through segmentation. The proposed methodology identifies the exact location of the crack in the image. A pre-processing step in the input enables such identification to increase the prominence of the cracks in the raw images. Thus, it facilitates the model's learning process. As a novelty, the work also brings a database of cracks in ceramics that can be used to improve future research.

This paper is organized as follows. We present related works in Section 2 and the proposed approach to address the segmentation problem in Section 3. In Section 4, the metrics, the loss function, and the experimental configuration are described, and we also present our set of images for segmentation of ceramic cracks. In Section 5, the results are described, and the experiments are discussed. Finally, in Section 6, we present our main conclusions and describe future research.

## 2. Theoretical Foundation

Image segmentation is a process that aims to divide images into regions or objects of interest that are homogeneous. This activity is the initial step in image processing applications, such as pattern recognition and image analysis. Image analysis includes characterization and representation of objects and measurement of resources. This process is mainly used to find objects and shapes [10]. There are currently several types of segmentation, usually based on formats, pixel characteristics, histograms, and movement. Each type supports common features in pixels or a group of pixels. [10,11]. The evolution of deep learning to various kinds of computer vision problems in the literature encourages this work to build a crack segmentation model based on deep learning.

IPTs had a significant advance in the last years. However, other problems have no solution found by IPTs yet, such as the real world's perception (lack of context of images, images with shadow, textures, variation in lighting), shading, and lighting variation. In parallel to this, there has been an exponential growth in the use of convolutional neural networks (CNN), and they have obtained better results for these problems. CNNs, too, have been used to classify cracks and fissures [8,9,12–15]. However, none of the proposed CNNs deals specifically with coating ceramics.

There are several studies in the literature on crack detection [8,9,11,12,16–18], but regarding cracking in ceramics there are few works and each surface has its specificity.

Young-Jin Cha in [8] applied a vision-based method using a deep architecture of convolutional neural networks (CNNs) to detect concrete cracks without calculations as defect characteristics. He aimed to create a model that could solve the problem without the use of processing image techniques. Moreover, Young-Jin Cha compared the obtained results with traditional methods based on edge detection. CNNs obtained the best performance, but the work is not related to identifying the location of the crack in the image, but in an evaluation of an intact or cracked part, which for a facade inspection is not ideal, and it is necessary to identify where the crack is located. In this work, it was resolved using segmentation, managing to extract the exact location of the fissure

In another study, Silva W.R.L [12] aimed to increase the level of automation in the inspection of concrete infrastructure when combined with unmanned aerial vehicles. The crack detection model developed is based on an image classification algorithm of the deep learning convolutional neural network (CNN). A relatively heterogeneous dataset has been provided. The authors claimed that deep learning allows the development of a concrete crack detection system responsible for several conditions, such as different light, surface finish, and humidity that a concrete surface can display. In this work, the model VGG16 [19] was used as a backend to the transference-learning technique. Silva's best experiment produced a model with an accuracy of 92.27%. However, Silva's work deals with image classification, stating whether or not there are cracks in concrete structures. Moreover, it does not make clear where the crack is located, which is essential for automated inspection of structures.

In Ahmed Mahgoub Ahmed Talab [18], the authors presented a new approach in image processing to detect cracks in the images of concrete structures. The method involves three steps. Firstly, changing the image to grayscale to use the Sobel method to detected edges and find the cracks. Second, determining an appropriate threshold in a binary image and classifying all pixels into two categories: background and foreground, and obtaining the region's area. Finally, using the filter area and changing the area if it is smaller than the specified number. Third, after applying the Sobel filter to eliminate residual noise, performing the Otsu method to detect large cracks. The article describes a method for detecting crack patterns in cement using image processing techniques. According to the author, this method's advantage is the precise and accurate detection of cracks in the images. The experimental work shows that the method is better than other widely used techniques. However, it does not use deep learning, and it is limited only to the use of image processing, which has the advantage of the low computational cost. Moreover, the work from Talab does not present the same generalization capacity as CNNs. In opposition, the methodology proposed in this work includes the combination of deep learning with image processing, improving the generalization in crack detection.

## 3. Crack Segmentation of Ceramic Surface

This work presents an architecture for segmenting cracks in facades with Deep Learning that was named the CCS model (Segmentation Model for cracks in ceramics) that includes a pre-processing step and a deep neural network for segmentation proposal followed by a threshold operation, as shown in Figure 1. The output is a binary image that brings white lines where the cracks were located, and, through overlapping images, it is possible to highlight the cracks in the original image. At CCS, pre-processing is done in the database before using the segmentation model. The pre-processed image with its label is adopted as input to perform the model's training. After training, only the original image is needed to run this network.

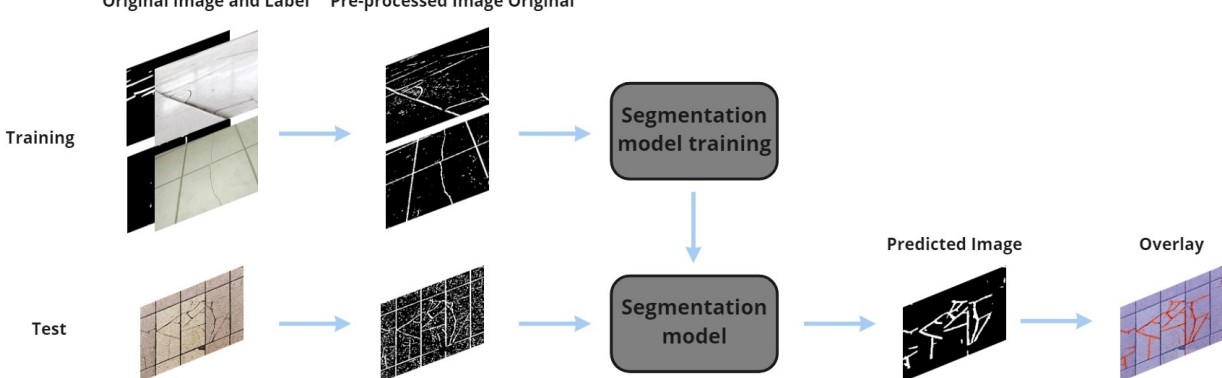

**Figure 1.** Architecture for crack segmentation in CCS model.

### 3.1. Data Pre-Processing

Pre-processing became necessary due to differences in context in the images, as just a grayscale image binarization is not enough, as much of the area of interest in the image is lost, and in some cases, the cracks present in the image disappearedm as shown in Figure 2. With that, it is necessary to apply some techniques in addition to binarization. Several experiments were carried out regarding the detection of lines, edges, and objects through computer vision to find a generic pre-processing for this problem.

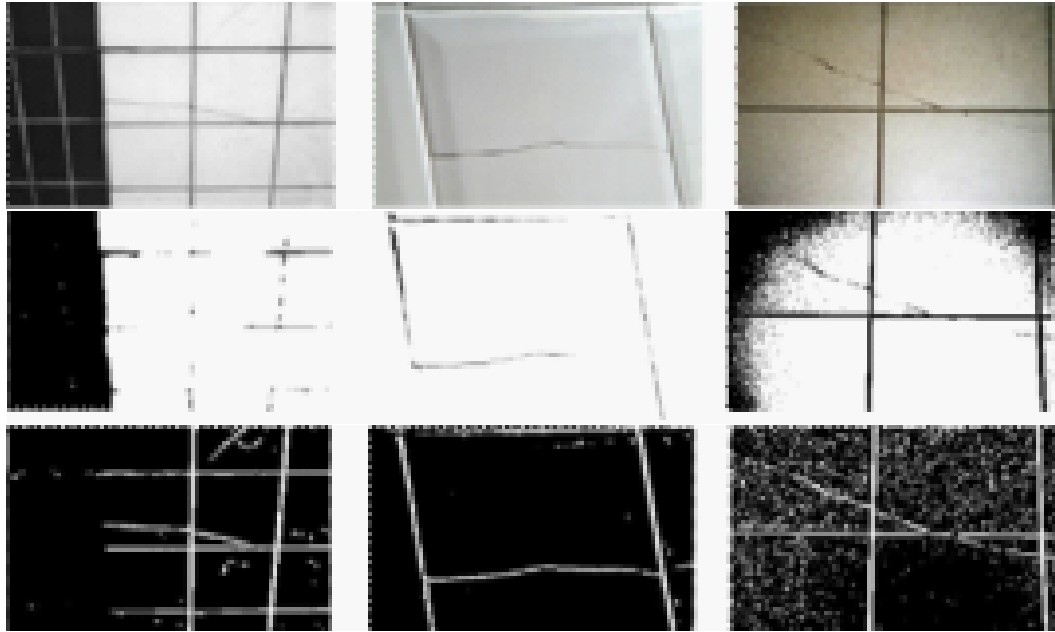

**Figure 2.** Example of a pre-procesing: (**up**) Original image, (**middle**) simple binarization, and (**down**) performed pre-processing on the original image.

The techniques used are listed below:

1. Histogram equalization;
2. Gaussian filter (with kernel 3 × 3);
3. Light and contrast adjust;
4. Inversion;
5. Erode and dilate functions (with kernel 5 × 5);
6. Finally, Otsu thresholding.

As seen in Figure 2, the pre-processing result highlights the area of interest, becoming much more in evidence, thus facilitating the neural network's performance in the extraction of characteristics.

This configuration was obtained through several attempts to highlight the images' cracks, using digital image processing techniques until reaching an acceptable result, where it obtained a better learning result by the model. The histogram equalization was used to change the image values' distribution, allowing the sharp differences to be reduced and accentuating details not previously visible. The Gaussian filter was used to soften the image, blurring it to remove noise, using a $3 \times 3$ kernel to make a smaller effect since the kernel's size influences the blurring power. Light and contrast adjustment was used to correct images with excessive lighting problems, not impacting those with standard lighting. The pixel inversion used in the images was necessary to comply with the standard established in the labeling of the images, where it was decided that everything white would be cracked surfaces and what was black would be parts of the ceramic, so the inversion made what was black turn white, and whatever was white turned black, since the grayscale highlighted the cracks in black. The erosion and dilation process was applied to solve the discontinuity of some cracks that broke during the blurring process, using a $5 \times 5$ kernel to continue the cracks, and lastly, a threshold was used. Some other techniques were tested but, in analysis, no significant change was seen for the objective that wanted to be achieved (highlighted in the fissures) and did not significantly influence the learning of the model.

### 3.2. Segmentation Model

The segmentation model uses the U-Net, proposed by Ronneberger et al. [20], which stands out in the segmentation problems due to the better performance, even with few images for training. The peculiar name of U-Net is due to the "U" shape of its architecture. The network input is the image that needs to be segmented. The output is the image label, a label that represents the model's expected output.

The network has a typical convolutional network architecture; however, it has two complementary paths, the contracting path (left side) and the expansive path (right side). The contracting path handles executing controls to extract characteristics from the image. This process reduces the dimensionality and increases the filters applied to extract features, generating a map for each level. On the other hand, the expansive path handles reducing the filters and increasing the dimensionality. A concatenation process is performed with the correspondingly cropped feature map from the contracting path to reach the segmented image's formation.

The contraction path is a typical convolutional network architecture. It contains nine learning convolutional layers and four max pooling operations after every three convolutions [20]. We applied two $3 \times 3$ convolutions in our implementations, each followed by a rectified linear unit (ReLU) and a $2 \times 2$ max pooling operation with stride 2 for downsampling. The number of feature channels is doubled for each downsampling step. A cropping process is made during the expansive path to avoid the loss of border pixels in every convolutional operation [20]. In our implementations, the feature map is halved by a $2 \times 2$ up-convolution [20] in an up-sampling process, followed by concatenation with the correspondingly cropped feature map from the contracting path, and two $3 \times 3$ convolutions. A ReLU operation follows each convolution. Cropping is necessary due to the loss of edge pixels in every convolution. A $1 \times 1$ convolution is used in the final layer to map each feature vector of the component to the desired number of classes.

### 3.3. Threshold

As a final step, it is necessary to apply a simple threshold to perform a binarization of the image and ensure that the end of the image values is of 0 (zero) or 1 (one), with 0 points for no cracks and 1 point for cracks in the semantic segmentation of pixel by pixel. For each pixel, the limit value used was 0.5, where, if the pixel value is less than the limit, it will be set to 0. Otherwise, it will be defined with the maximum value defined, in this case, 1.

## 4. Methodology

This section describes the methodology followed in this article. First, we describe the loss function used in the segmentation models. Next, we describe the database, metrics, and the experimental setup to obtain the results.

### 4.1. Loss Function

In this work, we used the Jaccard distance as our loss function. The Jaccard distance measures dissimilarity between sample sets. This function is complementary to the Jaccard coefficient or intersection over union. The loss function is calculated as:

$$L(A, B) = 1 - \frac{|A \cap B|}{|A \cup B|},$$

where $A$ and $B$, in the CCS model, are binary images of the same size.

### 4.2. Ceramic Cracks Database

We propose a ceramic crack database with 167 ceramic crack images. The images were collected by students of the University of Pernambuco from the civil engineering department. The database consists of images of a fixed resolution of $256 \times 256$ in RGB format without any pre-processing. Each element is labeled with a binary image of segmented cracks. The data has various characteristics, like different sizes, angles, illumination, distances, or even materials and textures. The database has images of building facades with ceramics with cracks of different shapes, both superficial and more profound. Besides, the database has images of ceramics with different colors and textures, which enrich its diversity and give more information to the model used. Figure 3 shows examples of the database images, the first line (up) shows the original images resize by $256 \times 256$, and the second line (down) shows the respective segmented label.

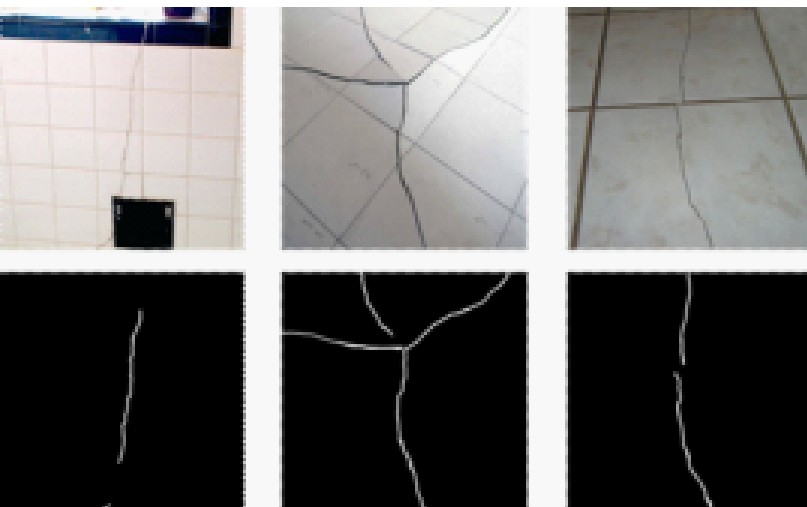

**Figure 3.** Examples of images collected to assemble the database and their respective ground truth (black and white).

A label corresponding to the images is required to perform the training of the segmentation networks. Those labels are the expected results of the network. They were manually generated for all the collected data and featured white for the original image regions characterized by a crack and black for all other regions. The database will be available for public use in future works related to the ceramic crack segmentation problem at the link https://github.com/gerivansantos/ceramic-cracks-dataset (accessed on 2 June 2021).

### 4.3. Metrics

The described metrics compare the previously mentioned approaches and evaluate which provided the best solution. For this purpose, the metrics selected are the Intersection over Union (IoU), Precision, Recall, the Kappa Coefficient, and Specificity.

#### 4.3.1. Intersection over Union

IoU or Jaccard coefficient is a measurement commonly used to validate semantic segmentation, and it is direct and effective. It is the intersection between predicted segmentation and ground truth divided by the union between the two, as demonstrated in Figure 4. This metric oscillates between 0 and 1, or 0 and 100%, wherein 0 indicates no intersection and 1 indicates an intersection equal to the union.

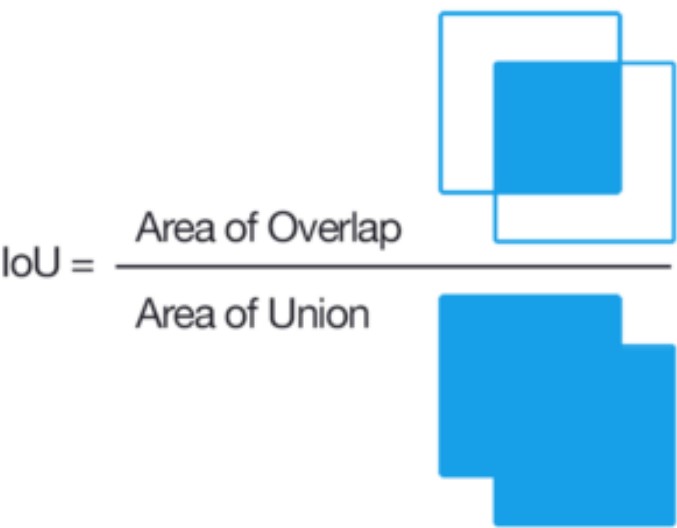

**Figure 4.** Representation of the Intersection over Union.

#### 4.3.2. Precision and Recall

Precision demonstrates the percentage of the relevance of the results. Conversely, recall illustrates the percentage of relevant results that are correctly classified by the algorithm. It relates the number of correct positive predictions to all positive predictions. The following equations calculate precision and recall:

$$Precision = \frac{TruePositives}{TruePositives + FalsePositives}$$

$$Recall = \frac{TruePositives}{TruePositives + FalseNegatives}$$

#### 4.3.3. Kappa Coefficient

The Kappa Coefficient is a statistic that evaluates the relation between two sets of data, calculated as follows:

$$k = \frac{P_0 - P_e}{1 - P_e},$$

where $P_0$ is the relative acceptance rate and $P_e$ is the hypothetical acceptance rate. Thus, the closer $k$ is to 1.0, the more the two data sets are related.

#### 4.3.4. Specificity

Specificity was also measured, and it is defined as the proportion of real negatives predicted to be negatives (True Negatives), as illustrated in the following equation. It implies another portion of real negatives that were predicted to be positives (False Positives),

which must equal 1 when summed with specificity. Another existing metric is sensitivity, which measures the proportion of correctly classified real negatives.

$$Specificity = \frac{TrueNegatives}{TrueNegatives + FalsePositives}$$

### 4.3.5. Confusion Matrix

The Confusion Matrix contains information on the real data and a classifying system's predictions, and it is commonly used to evaluate such a system. It is a table with four different relations between real and predicted values: True Positives (TP), correctly predicted positives; True Negative (TN), correctly predicted negatives; False Positives (FP), type 1 error, incorrectly predicted positives; False Negatives (FN), type 2 error, incorrectly predicted negatives. Thus, it is useful in measuring Precision, Recall, and Specificity.

### 4.4. Experimental Setup

This paper sets a benchmark over the proposed database, using state-of-the-art models. The 70% of the data is randomly allocated for training and 30% towards testing in the several experiments performed, which were used as comparative parameters for the database. The models selected to make a comparison with the proposed model were variations of implementations of U-Net [20] and LinkNet [21]. The criteria for its selection are the relevance in the image segmentation literature and the good accuracy in solving the proposed problem. Data augmentation was also applied to improve the generalization of the model.

The used architecture, backbone, and weight initialization method are illustrated in Figure 5. The backbone is the network architecture implemented in each model. In this work, we use the different architectures like resnet34 [22], resnet50 [22] and vgg16 [19], which are 34 layers deep, 50 layers deep and 16 layers of deep, respectively. The initialization of the weights is carried out randomly and using the weights of the pre-trained neural network with the ImageNet database [23].

Input images are set with size 224 × 224 to U-Net and LinkNet models. We apply the Adam algorithm, a stochastic gradient descent method based on the adaptive estimation of first-order and second-order moments [24]. In our experiments, we set the hyperparameters to the Adam algorithm with $\beta_1 = 0.9$, $\beta_2 = 0.999$, and $\epsilon = 10^{-07}$, with a learning rate of $\alpha = 0.001$.

A total of 20 experiments are performed 30 times to ensure that the results were statistically significant, and from them were extracted the metrics used to compare the results. Additionally, since fine-tuning presents good results in deep learning applications, we analyzed its efficiency in this work approach.

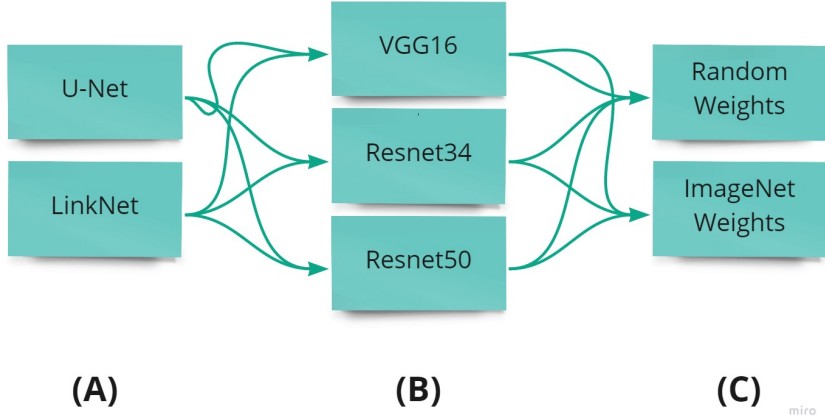

**Figure 5.** Conducted experiments, where (**A**) are models, (**B**) are backbones, and (**C**) are weight initialization, where "None" is the initialization of the weights carried out randomly.

## 5. Results and Discussions

The models were selected according to their success in the segmentation of images, as previously presented. After that, 20 fine-tuning experiments were performed, altering the models' backbone and the weight initialization. Table 1 outlines each model's three best results, utilizing the IoU metric.

**Table 1.** Result of the metrics for the U-Net and LinkNet models, and the different network architectures. The weights of a pre-trained network with ImageNet are used to initialize each of the models.

| Metrics | Models | | | | | | |
|---|---|---|---|---|---|---|---|
| | U-Net | | | | LinkNet | | |
| | CCS † | resnet50 † | resnet50 | resnet34 | resnet34 | resnet34 † | vgg16 |
| IoU | 0.865 | 0.685 | 0.681 | 0.675 | 0.697 | 0.684 | 0.672 |
| Precision | 0.999 | 0.713 | 0.727 | 0.720 | 0.727 | 0.711 | 0.704 |
| Recall | 0787 | 0.946 | 0.933 | 0.929 | 0.946 | 0.922 | 0.897 |
| Kappa | 0.724 | 0.805 | 0.808 | 0.803 | 0.814 | 0.794 | 0.775 |
| Specificity | 0.999 | 0.988 | 0.988 | 0.988 | 0.988 | 0.987 | 0.988 |

† For these models, weights were randomly initialized.

In terms of average IoU, our model CCS overcame the IoU value from the other models. We obtain an index of 86.5% in CCS model, and in the U-Net and LinkNet models, the value is around 68%. The obtained average precision in our approach in the U-Net model reaches a value of 99.9%. The U-Net and LinkNet models demonstrate that the resnet50 and resnet34 models, when initialized with the ImageNet weights, present an average precision of 72.7%. Regarding the average recall, to U-Net with resnet50 backbone, without weight initialization, and LinkNet with restnet34, initialized with the ImageNet weights, obtained percentages were equal to 94.6%. This value overcomes the average recall of our approach, with a value of 78.7%. Our approach shows a high value of accuracy and a low recall. However, most of our predicted labels are correct.

The best average kappa coefficient is obtained by a LinkNet model with a resnet34 backbone, using the ImageNet weights, with 81.4%. All the models to U-Net and LinkNet achieve a value of kappa coefficient around 80%. The lower value is from our approach, with 72.4%, followed by the LinkNet model with vgg16 and ImageNet initialization. We observe that the models with vgg16 architecture reach the low values of the kappa coefficient. The average value of specificity for the U-Net and LinkNet models reached a value of around 98%. Our approach obtains a value of 99.9%, overcoming the other values; nonetheless, it does not seem to show a significant difference.

Regarding the confusion matrix, our approach correctly classified 99% of the pixels that belong to the crack (Figure 6a). However, 27% of the pixels that are not from the crack are classified as part of it. Using the U-Net model, resnet50 with the random initialization, 95% of the pixels are correctly classified as cracks (Figure 6b). For all other implementations (Figure 6c,d), 93% of the crack is correctly classified. The Linknet model, with resnet34 and initialization with ImageNet, correctly classify 95% of the crack pixels (Figure 6e), surpassing the other implementations (Figure 6f,g) with values of 90% and 84%.

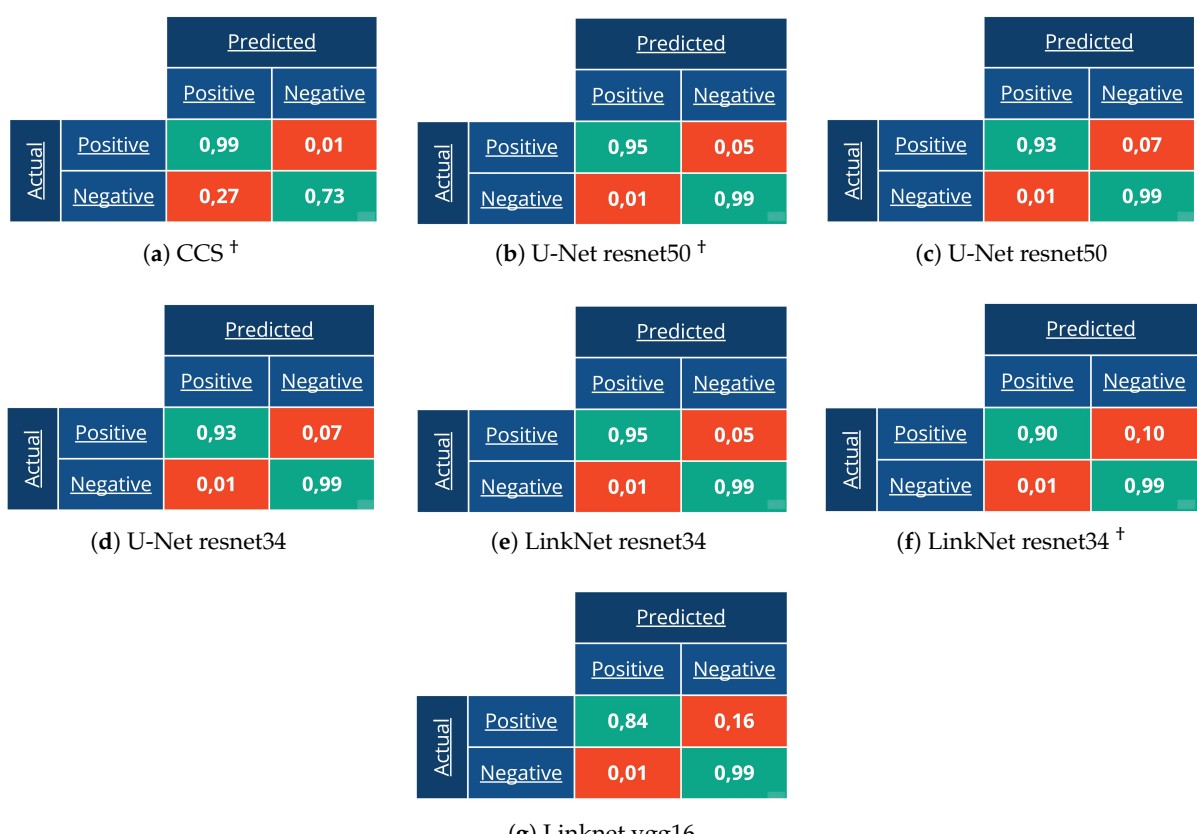

**Figure 6.** Confusion matrix from the prediction to (**a**–**d**) U-Net model, where (**a**) is CCS model, (**e**–**g**) LinkNet model. The weights of a pre-trained network with ImageNet are used to initialize each of the models (For the models [†], weights were randomly initialized).

Qualitative analysis can be illustrated with the results presented in Figure 7. The Figure shows an example image (Figure 7a) and its ground truth (Figure 7b). The original image is inputted to the network, having the expected output (Figure 7c–i). It is possible to observe mistakes in segmentation labeling in some regions. Figure 7c,h shows a thick segmentation in comparison with the other predictions, overestimating the region where the crack is. A fine segmentation can lose crack representation. In Figure 7e, some parts of the crack are not identified. It should be noted that the models above can segment the cracks that are above or near the grouts. In some IPTs, such as fissures, they are easily confused with grouts. During the training, some analyses were made, and it was observed that by increasing the number of epochs in the training, the model was able to learn more, including understanding when the cracks were overlapping in the grout reducing the problem of not identifying cracks near or overlapping the grout.

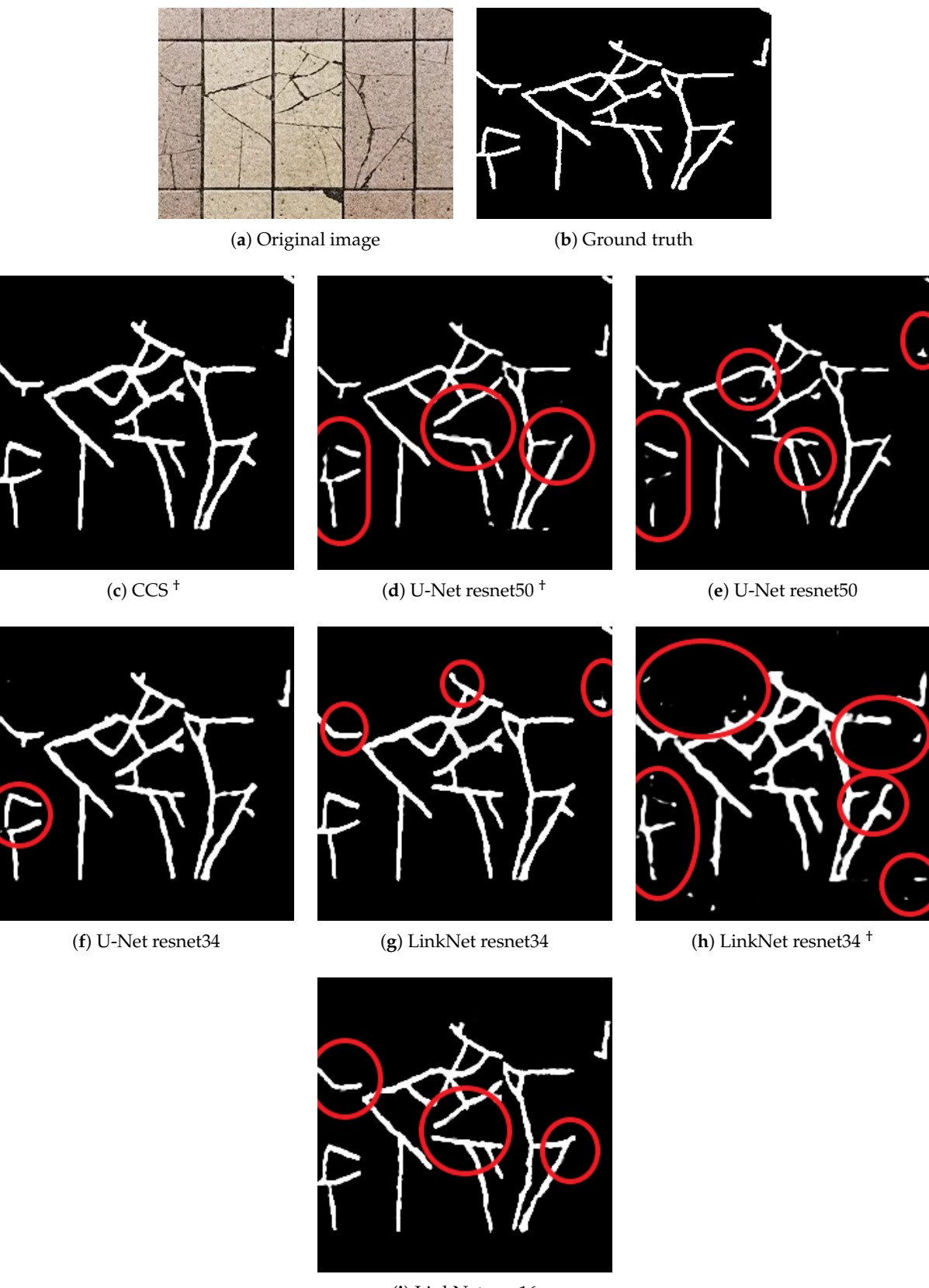

**Figure 7.** Example of (**a**) an original image and (**b**) the ground truth used for testing. Prediction of the (**c**–**f**) U-Net, where (**c**) is CCS, and (**g**,**h**) LinkNet models. The weights of a pre-trained network with ImageNet are used to initialize each of the models (For the models †, and weights were randomly initialized).

## 6. Conclusions

In this work, we analyze deep learning models' capabilities in the segmentation of cracks in ceramics tiles. We propose a pre-processing methodology to improve the performance of models to ceramic crack segmentation. Besides, we present the Ceramic Cracks database, a set of images with ceramic tiles with cracks destined for crack segmentation. Our results show that it is possible to identify cracks in ceramic images, although there are a few minor errors. The crack is identified with a high precision value in the model using a pre-processing methodology. The U-Net and LinkNet models achieve good results, using the resnet50 and resnet34 as backbones, respectively, and the weights of a pre-trained network with ImageNet to initialize. By increasing the number of epochs during training, the models manage to segment cracks even when they are in the tiles' grout.

Thus, other researchers can use the proposed study and database to improve their segmentation issues using computer vision. This paper then contributes to a new segmentation problem and a new database for crack segmentation in ceramic tiles. Future works are expected to compare the proposed solution's efficiency to other deep learning segmentation models and update the database, increasing the number of instances. We also intend to study the computational cost of the proposed solution and other solutions in the literature. On the other hand, we expect to implement this work in drones for optical facade inspection, which allows a more efficient inspection at a low cost.

**Author Contributions:** Conceptualization, G.S.J., R.D., A.C.J., and B.J.T.F.; data curation, G.S.J., R.D., A.C.J. and B.J.T.F.; formal analysis, G.S.J. and B.J.T.F.; funding acquisition, B.J.T.F.; investigation, G.S.J.; methodology, G.S.J., J.F. and C.M.-A. ; project administration, G.S.J., B.J.T.F. and A.C.J.; resources, G.S.J., B.J.T.F., A.C.J.; software, G.S.J. and J.F.; supervision, B.J.T.F. and A.C.J.; validation, G.S.J., J.F. and B.J.T.F.; visualization, G.S.J.; writing—original draft, G.S.J.; writing—review and editing, J.F., C.M.-A., R.D., A.C.J. and B.J.T.F. All authors have read and agreed to the published version of the manuscript.

**Funding:** This study was financed in part by the Coordenação de Aperfeiçoamento de Pessoal de Nível Superior-Brasil (CAPES)-Finance Code 001, and the Brazilian agencies FACEPE and CNPq.

**Conflicts of Interest:** The authors declare no conflict of interest.

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
