# Peer review of "Ceramic Cracks Segmentation with Deep Learning"

_applsci, doi:10.3390/app11136017_

Round 1

Reviewer 1 Report

This paper presents an automatic ceramic crack segmentation method using deep learning. While the proposed method can detect the crack in the ceramic tiles the reviewer believes the issues below need to be addressed to be published in this journal.

  1. What is the novelty of this paper? There are multiple crack detections method using CNN architectures. Please emphasize how the proposed method is different with other methods.
  2. The authors mentioned works by Young-jin Cha, Silva W.R.L, and Ahmed Mahgoub Ahmed Talab, and stated the limitations of their work. Did this paper resolved all these limitations? Please clearly state the problem and how the problems were resolved.
  3. Line 213: Please delete the word "previous".

Author Response

Thank you for allowing a resubmission of our manuscript, with an opportunity to address the reviewers’ comments.

We are uploading our point-by-point response to the comments (below) (response to reviewers).

Reviewer 2 Report

The article describes a laboratory test to assess the suitability of selected methods of deep learning and computer vision for monitoring the facade of a building. The authors laconically describe the method of selecting a classifier. The choice of pre-processing methods (as noted by the authors themselves) was made empirically. I do not know what the measure of quality was when selecting techniques and their order. If it was the subjective opinion of the observer (human), we do not know if it was the optimal choice for the classifier used. Technical Notes: Link from line 226 "https://github.com/gerivansantos/ceramic-cracks-dataset" is not working Figure 1 - Contains very small letters Figure 3 - Nothing important The articles end in 2017 on the literature list.

Author Response

(The authors gave the same response as above.)
